# Cell shape-independent FtsZ dynamics in synthetically remodeled bacterial cells

Bill Söderström [1], Alexander Badrutdinov[2], Helena Chan [1] & Ulf Skoglund[1]

FtsZ is the main regulator of bacterial cell division. It has been implicated in acting as a scaffolding protein for other division proteins, a force generator during constriction, and more recently, as an active regulator of septal cell wall production. FtsZ assembles into a heterogeneous structure coined the Z-ring due to its resemblance to a ring confined by the midcell geometry. Here, to establish a framework for examining geometrical influences on proper Z-ring assembly and dynamics, we sculpted *Escherichia coli* cells into unnatural shapes using division- and cell wall-specific inhibitors in a micro-fabrication scheme. This approach allowed us to examine FtsZ behavior in engineered Z-squares and Z-hearts. We use stimulated emission depletion (STED) nanoscopy to show that FtsZ clusters in sculpted cells maintain the same dimensions as their wild-type counterparts. Based on our results, we propose that the underlying membrane geometry is not a deciding factor for FtsZ cluster maintenance and dynamics in vivo.

[1] Structural Cellular Biology Unit, Okinawa Institute of Science and Technology, 904-0495 Okinawa, Japan. [2] Mechanical Engineering and Microfabrication Support Section, Okinawa Institute of Science and Technology, 904-0495 Okinawa, Japan. Correspondence and requests for materials should be addressed to B.Södöm. (email: Bill.soederstroem@oist.jp)

Most bacterial cells divide by binary fission, whereby one mother cell splits into two identical daughters[1–3]. Decades of study have led to a detailed understanding of how the cell division machinery, the divisome, carries out this task during the later stages of the cell cycle[4,5]. At the heart of this process is the eukaryotic tubulin homolog, FtsZ[6] that, together with its membrane anchors FtsA and ZipA (in *E. coli*), forms an intermediate structure called the proto-ring (Fig. 1a)[7]. Functioning as a recruitment base, the proto-ring components then enlist the remaining essential division proteins to form a mature divisome[5]. As soon as it is fully assembled, the divisome starts to constrict the cell envelope by reshaping the septal geometry, ultimately leading to sequential closure of the inner and outer membranes[8–10].

In rod-shaped model bacteria such as *Escherichia coli* and *Bacillus subtilis*, FtsZ is believed to organize into short bundles of filaments,

roughly 100 nm in length[11,12], that treadmill at the septum with a circumferential velocity in the order of 20–30 nm s$^{-1}$[13–15]. The treadmilling filaments guide and regulate septal peptidoglycan (PG-) production and ingrowth, leading to septation[16]. This mode of action may be limited to rod-shaped bacteria that have two separate PG-machineries, as opposed to cocci, which have only one PG-machinery that is capable of finalizing division in cells with inhibited FtsZ dynamics[17].

At a late stage of membrane constriction, but prior to inner membrane fusion, FtsZ disassembles from midcell, indicating the possible existence of an upper limit of ring curvature[6,9]. The actin homolog MreB is responsible for maintaining the rod shape of *E. coli*, and interacts directly with FtsZ, thus MreB could potentially play a role in FtsZ organization at the septum[18–20]. However, other purely geometrical constraints that might govern Z-ring

**Fig. 1** Midcell FtsZ-ring assembly is unaffected by increased cell diameter. **a** Simplified cartoon showing FtsZ treadmilling at the division plane of an *E. coli* cell. For clarity, only FtsZ (gray dots), it's membrane tethers, FtsA and ZipA (blue dots), and the membrane (brown) are shown. **b** Schematic representation of cell placement for imaging. Green dotted ring in the cells represents the FtsZ-ring (red arrow). Standing cells were trapped in a vertical position in micron-sized holes in agarose pads created using micron-sized pillars. Conditions for proper division ring placement are met when width < length. The left and middle cells represent untreated cells. The cell on the right has increased dimensions due to drug exposure (A22 and cephalexin). **c** Time-gated STED (gSTED) image of a typical FtsZ-ring (FtsZ-mNeonGreen) in an untreated standing cell. Scale bar = 1 μm. **d, e** gSTED images of FtsZ-mNeonGreen rings in *E. coli* cells treated with drugs, showing increased ring diameter. Scale bar = 1 μm. Drugs refer to A22 and cephalexin. **f** Close-up of representative FtsZ clusters shown in **e**, from a cell with increased diameter. Scale bar = 0.5 μm. **g** Quantification of FtsZ cluster lengths in untreated and drug-treated cells. Mean ± S.D. was 122.8 ± 43.9 nm (n = 77) and 132.4 ± 48.7 nm (n = 172) for untreated and drug-treated cells, respectively. No statistically significant difference was measured, p > 0.05. Inset shows cluster widths in untreated (mean ± S.D. = 86.1 ± 6.3 nm (n = 77)) and drug-treated cells (mean ± S.D. = 88.4 ± 9.8 nm (n = 172)). Boxes represent S.D., with red lines indicating mean. Whiskers on the box plots encompass 95.5% of the distribution. **h–k** Structured illumination microscopy (SIM) images of FtsZ-GFP in *E. coli* cells (**h**) untreated or (**i–k**) treated with drugs. Scale bars = 1 μm. **l** Snapshots of epifluorescence (EPI) images from time-lapse series of FtsZ-GFP dynamics in drug-treated cells. Scale bars = 1 μm. Corresponding kymographs are shown adjacent to each image. Black arrows point to examples of FtsZ trajectories. **m** Average treadmilling speed of FtsZ-GFP in untreated (mean ± S.D. = 26 ± 15 nm s$^{-1}$, n = 102) and drug-treated cells (mean ± S.D. = 30 ± 18 nm s$^{-1}$, n = 102). Black dots represent individual data points, bars represent mean with error bars representing S.D. "d" in **c–e**, and **h–l** indicates cell diameter

maintenance and stability are currently unclear. In vitro data have shown that FtsZ can self-organize into swirling rings on supported bilayers when unconfined by geometrical constraints[21], but we were curious as to whether geometrical changes to cell shape would influence Z-ring formation and dynamics, as this would result in a better understanding of FtsZ behavior in live cells. In this study, we systematically examine FtsZ formation, organization and behavior in E. coli cells that are sculpted into complex geometrical shapes in micron sized holes. We show that FtsZ formation and dynamics are independent of cell shape and membrane curvature.

## Results

**FtsZ structure and dynamics in Z-rings are not sensitive to increased ring size.** As a reference for unmodified division rings, we imaged Z-rings in E. coli cells expressing FtsZ-mNeonGreen as the only source of FtsZ[22]. Under our experimental conditions, this strain produced normal-looking, sharp Z-rings (Supplementary Figure 1) and grew and divided similarly to wild-type (WT) E. coli (MC4100) (Supplementary Figure 2a-e). We then trapped the cells in a vertical position in micron-sized holes that were produced in agarose pads using silica micron pillar arrays[14] (Fig. 1b, Supplementary Figure 3), and imaged the cells using super-resolution time-gated STimulated Emission Depletion (gSTED) nanoscopy. In these standing cells, a heterogeneous Z-ring with distinct FtsZ-mNeonGreen clusters was clearly seen traversing the circumference of the cell (Fig. 1c), similar to what has been observed before[12,14].

Previous work has shown that FtsZ clusters generally maintain the same length throughout envelope constriction[12,14]. We wanted to see if this was also true for unnaturally large cells, i.e., would FtsZ clusters maintain the same dimensions in Z-rings of cells with increased diameter at midcell? In order to increase cell diameter, we treated E. coli cells with A22 and cephalexin (hereafter collectively referred to as "drugs") in a way similar to what has previously proven successful for cell shape manipulations[23]. A22 disrupts MreB dynamics and therefore perturbs the characteristic rod-shape of E. coli cells[19,24], while cephalexin blocks cell division by inhibiting the transpeptidase activity of FtsI[25]. The net effect of this dual drug treatment is the growth of cells into shapeable blebs that are unable to divide (Supplementary Figure 4a).

We hypothesized that as long as cell width remains less than cell length, FtsZ molecules should be directed to midcell by the Min system[23] and other FtsZ placements systems[26], such that a ring-like structure may be observed in the xy-plane of vertically-oriented, standing cells (Fig. 1b). To test this, we exposed E. coli cells expressing FtsZ-mNeonGreen to drugs, and then trapped the cells vertically in holes with a diameter of up to 3.5 μm and a depth of 4.5–6 μm. It has been shown that reshaped E. coli cells longer than 6 μm may revert from pole-to-pole oscillations to high mode oscillations[23]. Depending on the size of the holes, cells were incubated between 120 and 240 min prior to imaging; over-incubation resulted in cells that outgrew the holes (Supplementary Figure 4b. Allowing cells to grow for longer times (>10 h) produced giant blobs with internalized FtsZ-mNeonGreen chains, see Supplementary Note 1, Supplementary Figures 5 and 6).

The resulting Z-rings in drug-treated cells spanned the midcell circumference for all cell diameters that were imaged (~ 1–3 μm) (Fig. 1d, e). Fluorescence intensity increased as ring size increased (Supplementary Figure 7), possibly indicating an upregulation of cellular FtsZ expression, assuming similar ratios of ring to non-ring associated FtsZ molecules in all cell sizes (~30% of FtsZ molecules are in the Z-ring[27]). Importantly, confocal Z-stacks showed that each cell contained only one Z-ring (Supplementary

Figure 8 and Supplementary Movie 1). Close inspection of STED images revealed that the Z-rings in larger cells were composed of fluorescent clusters (Fig. 1f) with average lengths and radial widths of $132 \pm 48$ nm and $88 \pm 9$ nm (mean ± S.D., $n = 172$), respectively, which were similar to Z-ring clusters in untreated cells ($p > 0.05$) (Fig. 1g).

After we had established that large Z-rings can form in cells with increased diameter, we proceeded to calculate FtsZ dynamics in these larger rings. However, strains expressing FtsZ-FP (fluorescent protein (FP), e.g., mNeonGreen) as the only source of FtsZ have been shown to have a phenotype similar to that of FtsZ mutants deficient in GTPase activity, with severely impaired treadmilling speed[13]. Therefore, we chose to image cells that expressed FtsZ-GFP from an ectopic locus on the chromosome, in addition to native FtsZ[28]. Earlier studies showed that FtsZ-GFP, when expressed at levels below 50% of total cellular FtsZ levels, caused no observable phenotypic changes[9,12,28,29]. In our experimental setup, FtsZ-GFP was expressed at ~ 30% of total FtsZ levels (Supplementary Figure 2).

SIM of drug-treated E. coli cells expressing FtsZ-GFP showed large heterogeneous rings that were similar to those of FtsZ-mNeonGreen (Fig. 1h–k). Time-lapse imaging revealed that FtsZ clusters moved around the midcell circumference, even in Z-rings with a diameter up to three times larger than that of a WT cell (Supplementary Movie 2). There was no difference in the speed of individual clusters in the rings of untreated cells compared to those in sculpted cells that had a diameter 50–200% larger than WT ($26 \pm 15$ nm s$^{-1}$ (mean ± S.D., $n = 102$) and $30 \pm 18$ nm s$^{-1}$ (mean ± S.D., $n = 102$), respectively) (Fig. 1l–m), suggesting that cluster treadmilling speed is not influenced by the length of the cell circumference. ZipA-GFP, an FtsZ membrane anchor, also moved at essentially the same speed in both normal-sized and large-sized rings ($26 \pm 8$ nm s$^{-1}$, mean ± S.D., $n = 10$) (Supplementary Figure 9 and Supplementary Movie 3), which is comparable to previously reported speeds[14].

Since treadmilling behavior of FtsZ in large cells was very similar to that in WT cells, we were curious to see whether FtsZ subunit exchange in the rings would also be similar. To assess this, we performed Fluorescence Recovery After Photobleaching (FRAP) experiments on both untreated and drug-treated cells. We bleached half of the FtsZ-GFP molecules in the rings of standing cells and monitored fluorescence recovery over time (Fig. 2a). Z-rings in untreated cells had a mean $t_{1/2}$ recovery time of $8.4 \pm 1.9$ s (mean ± S.D., $n = 23$) (Fig. 2b), consistent with previous studies[14,30]. We found that the average $t_{1/2}$ recovery time was the same for Z-rings with a wide range of diameters (Fig. 2b), indicating that the rate of FtsZ subunit exchange is independent of cell diameter. This further suggests that cell size, and hence membrane curvature, might not be a factor in determining Z-ring dynamics.

**The FtsZ-square.** Next, we wanted to know if drug-treated cells placed in deep (5 μm) rectangular volumes would adapt to these shapes and effectively form Z-rectangles or Z-squares instead of Z-rings. Previous work has shown that cells can adapt to rectangular shapes in shallow wells, approximately 1 μm deep[23]. Here, we produced quadrilateral patterns in agarose pads using silica micron pillar arrays similar to those previously described[14], with the exception that the pillars were rectangular and $5.5 \pm 0.5$ μm in height. Sides of the micron chambers were up to 3.5 μm in length (Supplementary Figure 11), resulting in well volumes up to 80 μm³, roughly 50-fold larger than the volume of a WT cell (assuming a WT cell size of 2 μm in length and 1 μm in width) (Supplementary Figure 12).

Drug-exposed cells expressing FtsZ-mNeonGreen were placed in rectangular micron holes and incubated at room temperature

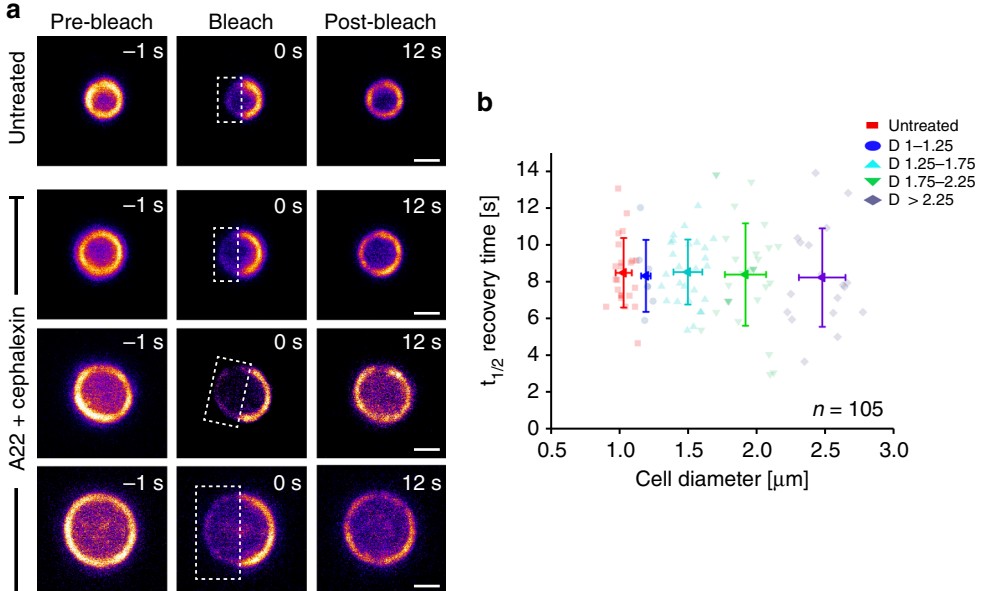

**Fig. 2** Cell size independent recovery of fluorescence in FtsZ-rings. FRAP measurements on FtsZ-GFP rings in *E. coli* cells trapped standing in a vertical position. **a** Representative cells of different diameter, untreated or treated with drugs. White boxes indicate bleached areas. Scale bars = 1 μm. **b** Quantification of FRAP data from untreated cells (red), and drug-treated cells (other colors), showing that fluorescence recovery time is independent of cell diameter (in the range investigated, i.e., ~ 1–3 μm). $n_{tot} = 105$. D = cell diameter range. Mean $t_{1/2}$ recovery time WT FtsZ-rings was 8.4 ± 1.9 s (mean ± S.D., $n = 23$, red squares), while for large FtsZ-rings, recovery times were 8.2 ± 1.9 s ($n = 7$, dark blue circles), 8.4 ± 1.7 s ($n = 27$, light blue triangles), 8.3 ± 2.8 s ($n = 30$, green triangles) and 8.2 ± 2.6 s ($n = 18$, purple diamond) for cells with diameters in the ranges of 1–1.25 μm, 1.25–1.75 μm, 1.75–2.25 μm and > 2.25 μm, respectively. Error bars represent S.D. Note that FRAP measurements on fixed cells indicated that trapped cells were stationary, and not rotating in the holes (Supplementary Figure 10)

for 300–420 min (longer incubation times were needed due to increased well size). The cells adapted to their new shapes and formed rectangular cuboids with only one Z-square per cell (Fig. 3a, Supplementary Movie 4). Notably, FtsZ clusters were observed both in the sharp corners and along the sides of the rectangles (Fig. 3b, Supplementary Figure 13). Quantification of the FtsZ-mNeonGreen clusters showed that they had similar dimensions to those in untreated cells, with an average length of 105.4 ± 39.6 nm and radial width of 79.6 ± 18.2 nm ($n = 147$) (Fig. 3c). This suggests that FtsZ cluster dimensions in vivo are insensitive to membrane curvature (or lack thereof).

To generate a fluorescent FtsZ fusion protein that could be used for both super-resolution STED imaging and examination of cluster dynamics when grown in rich media at 37 °C, we constructed a plasmid-expressed FtsZ-mCitrine fusion. FtsZ-mCitrine was expressed from an IPTG-inducible, medium copy-number plasmid, pTrc99a, at a level approximately equal to 30% of total cellular FtsZ. Under these conditions, FtsZ-mCitrine formed normal-looking, sharp Z-rings (Supplementary Figures 1 and 2). Cells expressing FtsZ-mCitrine were then exposed to drugs, trapped in rectangular micron-sized holes, and incubated for 180–280 min at room temperature before gSTED imaging. We found that FtsZ-mCitrine formed clusters that were 118.3 ± 41.3 nm long and 86.3 ± 22.5 nm wide radially ($n = 162$), similar to FtsZ-mNeonGreen cluster dimensions (Fig. 3c, Supplementary Figure 14), indicating that fluorophore choice did not influence cluster dimensions in the rings. For consistency, we also imaged rectangular cells expressing FtsZ-GFP from the chromosome using SIM (Supplementary Figure 14). All three strains tested adapted to the rectangular shape, producing sharp-cornered Z-rectangles.

**FtsZ dynamics in rectangular-shaped cells.** In order to examine the dynamics of FtsZ in rectangular cells, we performed time-lapse

imaging on cells expressing either FtsZ-mCitrine or FtsZ-GFP. Although a few fluorescence spots were abnormally bright and immobile (~ 1 spot/5 cells, with a maximum of 2 spots in one cell) (Fig. 4b, Supplementary Movie 7, red arrow), the majority of FtsZ clusters were highly dynamic (Fig. 4a, b, Supplementary Movies 5–6). Note that the bright, immobile spots were excluded from treadmilling analyses. Close inspection of time-lapse sequences suggested that FtsZ clusters in rectangular-shaped cells could treadmill continuously around the perimeter of the cells (Fig. 4a), and importantly, even across sharp corners, without an apparent change in speed (Fig. 4b, c, Supplementary Movie 8). The average treadmilling speed of FtsZ-mCitrine clusters in rectangular cells with perimeter lengths up to 13 μm (more than four times the circumference of a WT cell) was 27.6 ± 12.5 nm s$^{-1}$ ($n = 109$), which was consistent with the measured treadmilling speed of FtsZ-GFP in rectangular cells (25.3 ± 11.3 nm s$^{-1}$, $n = 122$) (Fig. 4d), large cylindrical cells (30 ± 18 nm s$^{-1}$, Fig. 1m) and untreated cells (~25 nm s$^{-1}$)[13,14].

To determine whether the dynamics of FtsZ subunit exchange are affected by changes to circumferential length and shape, we collected FRAP measurements on FtsZ bundles in rectangular-shaped cells (Fig. 4e, Supplementary Movie 9). The recovery times of half-bleached rectangles of varying sizes matched those of rings, with mean $t_{1/2}$ recovery times of 9.85 ± 2.58 s ($n = 24$) and 9.15 ± 2.55 s ($n = 22$) for FtsZ-mCitrine and FtsZ-GFP, respectively (Fig. 4f). This suggests that subunit exchange from the cytoplasmic FtsZ pool is independent of circumference length and membrane curvature. The data thus far indicate that the maintenance and dynamics of FtsZ clusters are preserved in both large Z-rings and Z-rectangles of varying size.

**FtsZ dimensions and dynamics in heart-shaped cells.** To examine whether FtsZ could literally be (at) the heart of cell division, we engineered micron pillar arrays that were

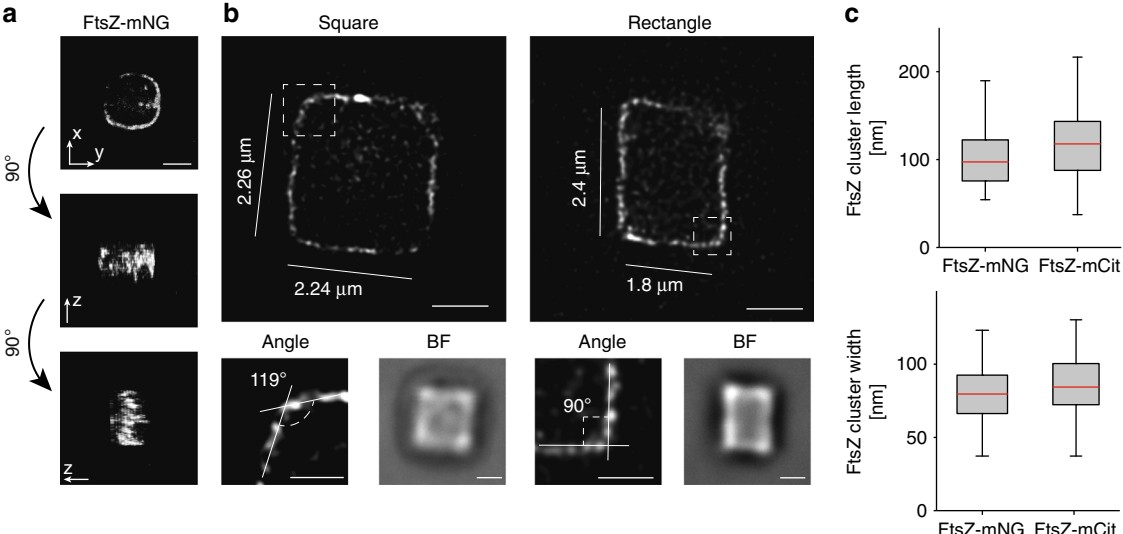

**Fig. 3** FtsZ-squares and FtsZ-rectangles in shaped cells. Drug-treated (A22 and cephalexin) *E. coli* cells expressing FtsZ-mNeonGreen were sculpted into rectangular shapes and imaged using super-resolution STED nanoscopy. **a** 3D rendering of a confocal Z-stack of an FtsZ-mNeonGreen square, showing only one band of FtsZ. Note that information along the *z*-axis is elongated. **b** Representative STED images of FtsZ-mNeonGreen in square (left) and rectangular (right) cells with perimeters ranging from 8.4 to 11.52 μm (compared to WT ~ 3 μm). Additional examples are provided in Supplementary Figure 13. Close-up images show representative corner angles. BF, brightfield image of corresponding cells. **c** Quantification of FtsZ cluster dimensions, showing little difference between FtsZ-mNeonGreen (105.4 ± 39.6 nm, 79.6 ± 18.2 nm; mean ± S.D., length and width, respectively. *n* = 147) and FtsZ-mCitrine (118.3 ± 41.3 nm, 86.3 ± 22.5 nm; length and width, respectively. *n* = 162. Example images of FtsZ-mCitrine squares are shown in Supplementary Figure 14). Scale bars = 1 μm. Boxes represent S.D., with red lines indicating mean. Whiskers on the box plots encompass 95.5% of the distribution

heart-shaped (Supplementary Figure 15). Heart shapes were chosen because they would sculpt cells in such a way that highly curved, straight, and angled membrane segments would be present within a single cell. Drug-treated *E. coli* cells expressing cytoplasmic GFP, FtsZ-mNeonGreen or FtsZ-mCitrine were sculpted into hearts as described above (Fig. 5a). Perhaps not surprisingly, quantification of 155 individual FtsZ clusters from the heart-shaped cells revealed dimensions similar to those in round and rectangular cells (129 ± 44 nm long and 84 ± 9 nm wide) (Fig. 5b). We also found that the average treadmilling speed of FtsZ-mCitrine in heart-shaped cells (22 ± 10 nm s$^{-1}$, *n* = 44) was essentially the same as that in untreated cells (Fig. 5c, Supplementary Movie 10).

For about one-third of the heart-shaped cells, we noticed bright spots of internalized FtsZ-FP signal that accumulated close to the cell center (Fig. 5c, green arrowhead). Although we couldn't distinguish whether these were true FtsZ clusters or aggregated protein, cytoplasmic clustering of FtsZ in WT cells have previously been reported[12]. Furthermore, although most hearts had FtsZ-FP signal spanning the full perimeter of the cell, approximately 20% were only half-full (Fig. 5d, lower left). We do not fully understand the underlying reason for this, however it is unlikely due to image focus or cell tilt issues, as every cell was scanned in the z-direction prior to imaging. Nevertheless, when we subjected the heart-shaped cells to FRAP, fluorescence recovery rates were equal for both full and half-full hearts (Fig. 5d), with mean $t_{1/2}$ recovery times of 7.1 ± 1.1 s (*n* = 24) and 6.9 ± 0.9 s (*n* = 9), respectively (Fig. 5e).

**FtsZ-rings form in complex cell shapes**. To explore if cell geometry plays a role in Z-ring formation, we set out to remodel cells into other complex shapes. Even though highly complex-shaped bacteria occur in nature, such as star-shaped bacteria[31], we wanted to test whether rod-shaped *E. coli* cells would allow themselves to be drastically remodeled. Using micron pillars of various shapes, we produced holes in agarose pads such that

drug-exposed cells could be sculpted into complex shapes, such as pentagons, half-moons, stars, triangles and crosses (Fig. 6a, middle row. Supplementary Figure 15). The cells conformed remarkably well to these shapes, forming sharp boundary angles < 70° (Fig. 6a, star). After we confirmed that cells could adapt to these complex shapes, we placed cells expressing FtsZ-mCitrine into the micron holes, allowed for reshaping to occur, and then imaged the cells using STED nanoscopy. Cells of all tested shapes produced FtsZ-shapes at midcell (Fig. 6a, bottom row). Subsequent analysis of the lengths and widths of the FtsZ clusters revealed little difference in dimensions between the different shapes, suggesting a minimal role of cell shape in determining FtsZ cluster dimensions in vivo (Fig. 6b). Additionally, time-lapse imaging of cells expressing FtsZ-mCitrine in various shapes showed similar dynamics to those measured in untreated cells. Specifically, FtsZ clusters treadmilled continuously over sharp corners and severe angles with an average speed of 28.7 ± 11.1 nm s$^{-1}$ (Supplementary Figure 16 and Supplementary Movie 10), which is similar to the treadmilling speed in both square-shaped and WT cells.

## Discussion

Cells, both bacterial and eukaryotic, have the ability to adapt to their local environments[32–36], reverting to their original shapes after stress[37,38] and dividing with striking midcell accuracy even when remodeled into irregular cell shapes[32,35]. In bacteria, the tubulin homolog FtsZ assembles into a ring-like structure at midcell and is responsible for overall maintenance of the cell division machinery[5,6]. The general dynamics and organization of the FtsZ-ring have been shown to be quite similar across many bacterial species[11,13–15,17,39–42]. Common to these species is confinement of the FtsZ-ring to a circular geometry at midcell. Strikingly, when purified FtsZ is placed on supported lipid bilayers, it assembles into a dynamic, swirling ring-like assembly with a diameter resembling that of wild-type *E. coli* cells (approximately 1 μm). This phenomenon is observed when FtsZ

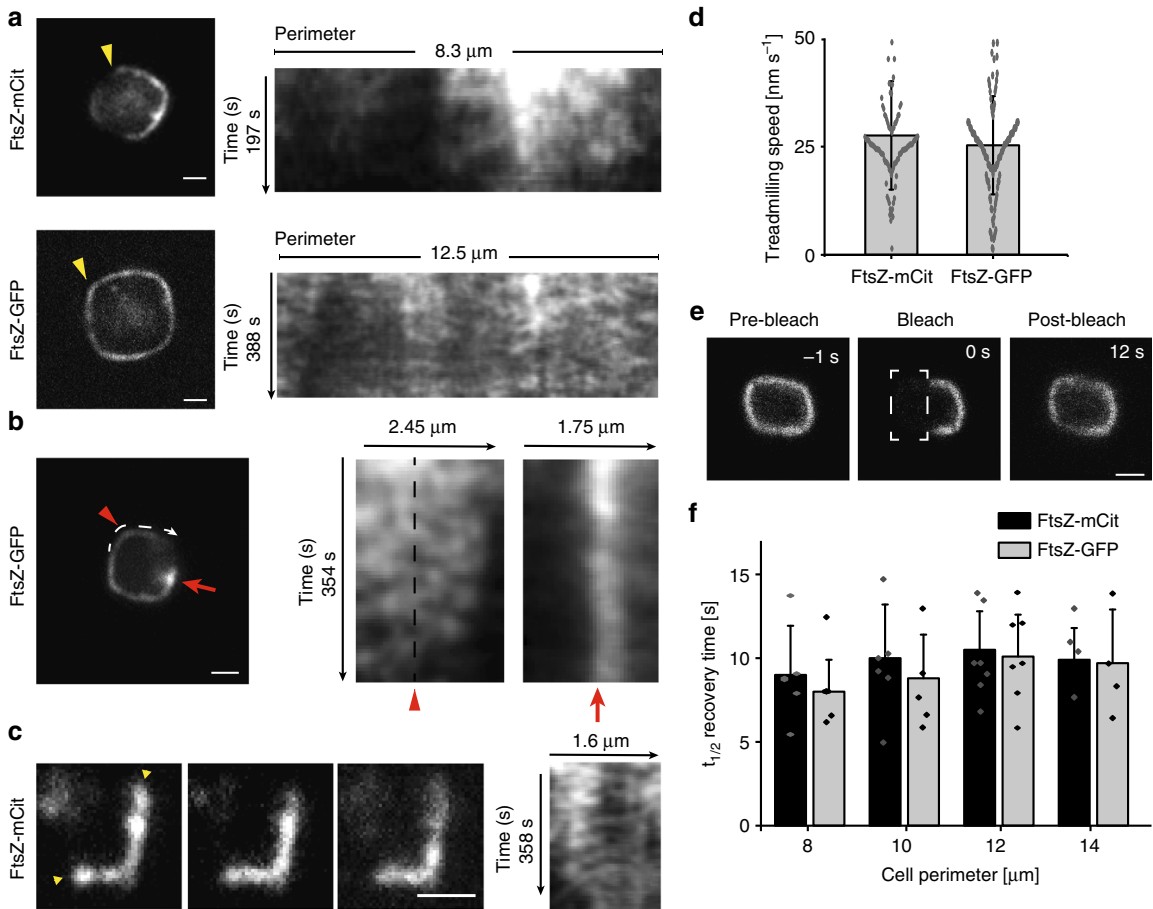

**Fig. 4** FtsZ dynamics in rectangular-shaped cells. The dynamics of FtsZ in rectangular shapes were assessed by time-lapse imaging and FRAP measurements on *E. coli* cells expressing FtsZ-mCitrine (FtsZ-mCit) or FtsZ-GFP. **a–c** Snapshot images from time-lapse series of FtsZ-mCitrine or FtsZ-GFP in rectangular shaped cells. Corresponding kymographs are shown next to each image. **a** Kymographs were taken around the entire perimeter (starting at the upper left corner, moving counter-clockwise, indicated by the yellow arrowheads). **b** Kymographs were taken along the white line (left kymograph), or over the bright spot indicated by the red arrow (right kymograph). FtsZ can clearly be seen treadmilling continuously across the sharp corner (indicated by the red arrowhead). The red arrows and arrowheads in the images correspond to the arrows and arrowheads on the kymographs. The black dashed line in **b** indicates the upper left corner of the cell. **c** Kymograph taken between the yellow arrowheads (top to bottom is left to right in the kymograph). **d** Average treadmilling speed of FtsZ-mCitrine and FtsZ-GFP in rectangles was $27.6 \pm 12.5$ nm s$^{-1}$ ($n = 97$) and $25.3 \pm 11.3$ nm s$^{-1}$ ($n = 122$), respectively. **e** Typical FRAP measurement of FtsZ-GFP in a rectangular *E. coli* cell. Half of the rectangle was bleached. **f** Average recovery times for FtsZ-mCitrine (dark, $n_{tot} = 24$) and FtsZ-GFP (light, $n_{tot} = 22$) in FtsZ-rectangles of various perimeter lengths. FRAP recovery times for rectangular cells with different perimeters: FtsZ-mCitrine $t_{1/2}$ recovery times: $9 \pm 2.9$ s (Circ. $8 \pm 1$ μm, $n = 6$), $10 \pm 3.2$ s (Circ. $10 \pm 1$ μm, $n = 7$), $10.4 \pm 2.3$ s (Circ. $12 \pm 1$ μm, $n = 7$), $9.9 \pm 1.9$ s (Circ. $14 \pm 1$ μm, $n = 4$). FtsZ-GFP $t_{1/2}$ recovery times: $8.1 \pm 1.9$ s (Circ. $8 \pm 1$ μm, $n = 5$), $8.8 \pm 2.6$ s (Circ. $10 \pm 1$ μm, $n = 5$), $10.1 \pm 2.5$ s (Circ. $12 \pm 1$ μm, $n = 8$), $9.7 \pm 3.2$ s (Circ. $14 \pm 1$ μm, $n = 4$). Circ. = Cell circumference. Values represent mean ± S.D. Dots represent individual data points, bars represent mean with error bars representing S.D. Scale bars = 1 μm

is anchored to the lipid bilayer, either by FtsA or a membrane targeting sequence, hinting at an intrinsically preferred FtsZ-ring curvature[6,21,43].

In this study, we characterized FtsZ midcell accumulation and dynamics in cell shape-determining environments by 'looking through the Z-ring' along the long-axis of cells. We observed normal-looking FtsZ-rings in cells with diameters three times the size found in WT cells. However, this might not be surprising, as the total intensity fluorescence increased in large cells (Supplementary Figure 7) and considering only ~30% of the pool of FtsZ molecules are in the ring of WT cells at any given point in time[27]. Quantification of FtsZ cluster dimensions revealed little variation between different cell shapes, such as squares, pentagons, triangles and stars (on average $123 \times 80$ nm, length × width, respectively, and summarized in Table 1), suggesting that local membrane geometry has minimal influence on FtsZ cluster dimensions. Compared to untreated cells, rectangular and heart-

shaped cells with perimeter lengths more than four times that of a WT cell exhibited similar FtsZ cluster dynamics, as FtsZ-FP clusters treadmilled continuously at the same average velocity throughout the perimeter of the shaped cells (including over sharp corners and regions with severe angles), and FtsZ subunit exchange occurred at similar rates, independent of cell shape and size (Table 2).

In summary, our results from different shaped cells show that Z-ring formation and dynamics are not limited to cells of a certain shape or size. This agrees with previous findings, which show that internal cellular structures are maintained in cells that have been reshaped into unnatural forms[23]. Our observation that FtsZ clusters conform to the geometric shape of the membrane at midcell suggests that FtsZ-ring formation is not affected by changes in membrane curvature. Indeed, cell shape and size are important for proper cellular functions[44], however, with the many naturally-occurring shape variations of bacteria[31,45], it is

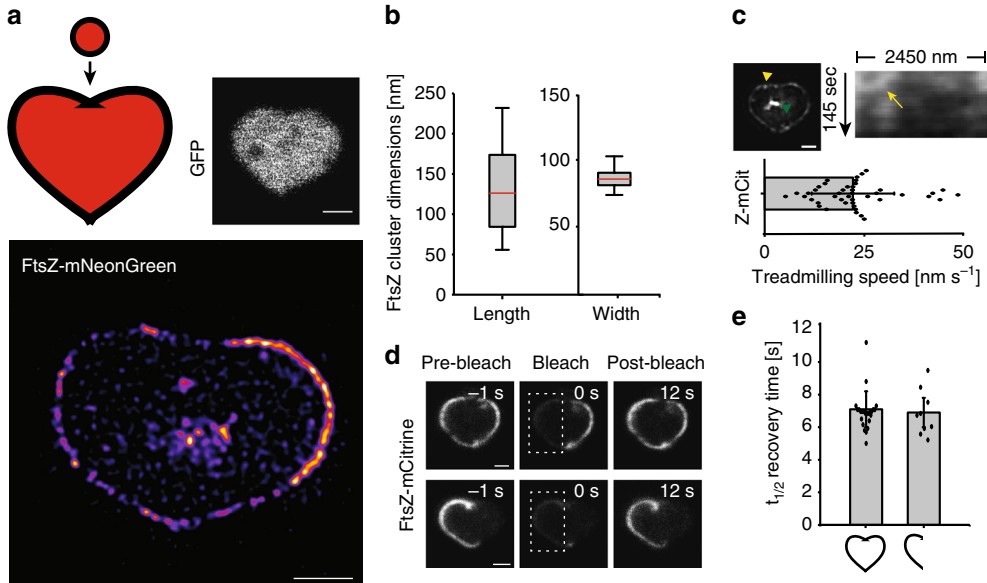

**Fig. 5** FtsZ cluster dimensions and dynamics in heart-shaped cells. FtsZ behavior in *E. coli* cells sculpted into heart shapes. **a** Upper left, Cartoon representation of a WT *E. coli* cell and a heart shape (both colored red for visualization), highlighting the large and complex structural changes of a cell-to-heart transition, approximately to scale. Upper right, Drug-treated cell expressing cytoplasmic GFP, shaped as a heart. Lower, STED image of an FtsZ-heart (FtsZ-mNeonGreen) in a drug-treated *E. coli* cell. **b** Lengths and widths of 155 individual FtsZ-mNeonGreen fluorescence clusters in cells shaped as hearts. Average length = 129 ± 44 nm and width = 84 ± 9 nm. Boxes represent S.D., with red lines indicating mean. Whiskers on the box plots encompass 95.5% of the distribution. **c** Upper row, SIM image from a time-lapse series (epi-fluorescence) of a heart-shaped cell expressing FtsZ-mCitrine. Green arrowhead indicates internal FtsZ clustering. Corresponding kymograph is shown adjacent to the image, and was generated starting at the yellow arrowhead in the SIM image, moving counter-clockwise for the indicated length. The yellow arrow points to an FtsZ trajectory. Lower, average treadmilling speed of FtsZ-mCitrine (Z-mCit) clusters in hearts (22.6 ± 10.4 nm s$^{-1}$, n = 44). **d** FRAP measurements of FtsZ-mCitrine in heart-shaped cells. Top row, bleaching of half the FtsZ-mCitrine molecules in a full heart. Bottom row, bleaching of a half-full heart. No difference in recovery times was observed. **e** Histogram of average $t_{1/2}$ recovery times calculated from FRAP measurements. Recovery in full hearts: 7.1 ± 1.1 s (n = 24), recovery in half-hearts: 6.9 ± 0.9 s (n = 9). Scale bars = 1 μm. Dots represent individual data points, bars represent mean with error bars representing S.D

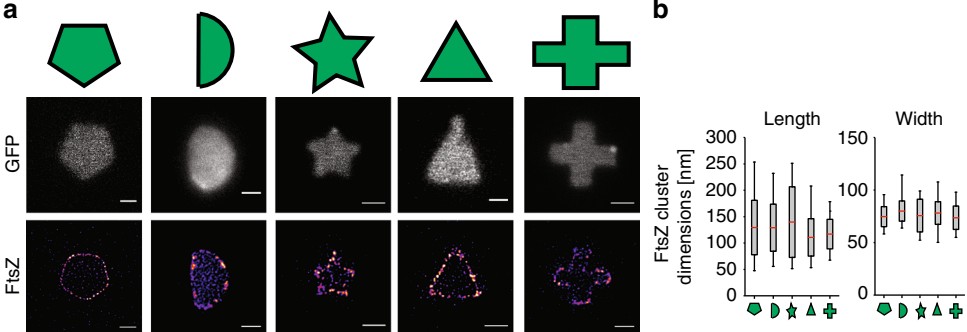

**Fig. 6** FtsZ bundle dimensions in complex shapes. **a** Cells expressing cytosolic GFP or FtsZ-mCitrine were remodeled into various shapes. Top row, schematic representation of the cell shapes (colored green for visualization). Middle row, representative cells expressing cytosolic GFP, and sculpted in the corresponding shapes. Bottom row, an FtsZ-pentagon, FtsZ-half-moon, FtsZ-star, FtsZ-triangle and FtsZ-cross in sculpted cells. Scale bars = 1 μm. **b** Quantification of FtsZ cluster lengths and widths in shaped cells. Cluster dimensions in pentagons (l = 131.4 ± 52 nm, w = 74.6 ± 9.5 nm, n = 121), half-moons (l = 129 ± 45 nm, w = 79.9 ± 9.6 nm, n = 98), stars (l = 139.6 ± 67 nm, w = 75.5 ± 15.5 nm, n = 113), triangles (l = 110 ± 35 nm, w = 78 ± 10.7 nm, n = 81) and crosses (l = 118.9 ± 21 nm, w = 71.1 ± 17.8 nm, n = 45). l, length, w, width. Boxes represent S.D., with red lines indicating mean. Whiskers on the box plots encompass 95.5% of the distribution

perhaps not surprising that FtsZ can adapt to changing environments without compromising its own ability to maintain fundamental functionality. Although our data do not explicitly show that sculpted cells can divide (since downstream division proteins were inhibited), the fact that the dynamic properties of FtsZ were conserved in these cells shows that the Z-ring can be decoupled from the constriction process. Presently, we have shown in vivo that *E. coli* FtsZ-ring formation and dynamics are conserved, irrespective of cell shape and size.

## Methods

**Bacterial growth**. All experiments were performed in *E. coli* strain MC4100, unless otherwise stated. Pre-cultures were grown overnight in 20 ml of rich media (LB) at 37 °C or M9 minimal media supplemented with 1 μg ml$^{-1}$ thiamine, 0.2% (w/v) glucose and 0.1% (w/v) casamino acids. The following morning, cultures were back-diluted 1:50 in either LB or M9 (with supplements) and antibiotics (ampicillin 25 μg ml$^{-1}$) when needed, and incubated at 30 or 37 °C.

**Fluorescent protein production**. Chromosomally-encoded FtsZ-mNeonGreen was integrated at the native *ftsZ* locus and did not require any inducer[22].

Chromosomally-encoded FtsZ-GFP (strain BS001), GFP[CYTO] (strain BS008) and ZipA-GFP were induced with 2.5 µM, 5 µM and 50 µM IPTG, respectively[9].

The plasmid pHC054 (*ftsZ-mCitrine*) was constructed using Gibson assembly[46] to generate an IPTG-inducible FtsZ-mCitrine fusion expressed from pTrc99a[47]. PCR was performed using Q5 High-Fidelity DNA polymerase (New England Biolabs). A DNA fragment containing *ftsZ* was amplified from *E. coli* MC4100 genomic DNA using primers FtsZ(F) (5′-caatttcacacaggaaacagaccatggatgtttgaacc aatggaac-3′) and FtsZ(R) (5′-gcccttgctcaccatctgcaggttgttgttatcagcttgcttacgcagg-3′). *mCitrine* was amplified from mCitrine-N1 plasmid DNA using primers mCitrine (F) (5′-cgtaagcaagctgataacaacaacctgcagatggtgagcaagggcgaggag-3′) and mCitrine(R) (5′-ccgccaaaacagccaagctttttacttgtacagctccatgtccatgc-3). pTrc99a plasmid DNA was amplified using primers pTrc99a(F) (5′-ccatggtctgtttcctgtgtg-3′) and pTrc99a(R) (5′-aagcttggctgtgttttggcgg-3′). The *ftsZ* and *mCitrine* coding regions are separated by a short linker encoding NNNLQ. The plasmid sequence was verified by DNA sequencing (Fasmac, Japan). FtsZ-mCitrine expression was induced with 2.5 µM IPTG. All FtsZ levels were quantified using Western blotting.

**Western blot analysis.** Cell extracts from a volume corresponding to 0.1 OD$_{600}$ units were collected for each strain to be analyzed. The extracts were suspended in loading buffer and resolved by SDS-PAGE gel electrophoresis. Proteins were transferred to nitrocellulose membranes using a semi-dry Transfer-Blot apparatus (Bio-Rad). The membranes were blocked with 5% (w/v) milk and probed with antisera to FtsZ (Agrisera, Sweden) and detected using standard methods.

**Nanofabrication of micro arrays.** Micron pillars were engineered using two different, but related, approaches. The first approach was used for round and square/ rectangular micron pillars, and was adapted from the refs. [14,15]. Using a multi-step process similar to that described previously[48], micron-scale pillars were fabricated on a silicon (Si) substrate by reactive ion etching. A pattern of hard-baked photoresist was created on a Si surface using UV lithography, to work as a mask for etching. Subsequent etching was performed using an Oxford Plasmalab100 ICP180 CVD/Etch system, with a mixture of SF$_6$ and O$_2$ plasma as an etchant. For our process, a SF$_6$:O$_2$ ratio of 1:1 was optimal. After etching, the remaining photoresist was removed by O$_2$ plasma treatment. Pillar arrays ($1 \times 1$ cm or $2 \times 2$ cm) with round pillars were engineered to contain one micron-sized pillar every 5 µm, with dimensions between 0.9 and 3.5 µm wide and $5.25 \pm 0.75$ µm high (Supplementary

Figure 3). Pillar arrays ($1 \times 1$ cm) with square pillars contained micron-sized pillars approximately every 5 µm, with side lengths varying between 1.8 and 3.5 µm, and heights of $5.5 \pm 0.5$ µm (Supplementary Figure 11).

To create more complex shapes, a second approach, based on electron beam lithography was used. For this, the micron-scale structures were fabricated on a Si substrate by a multi-step process, which was a combination of electron beam lithography and reactive ion etching techniques. Similar approaches to silicon patterning have been successfully used in a number of earlier works[48–51]. First, a pattern of e-beam resist was created on a Si surface using e-beam lithography. A 50 nm-thick Ti layer was then deposited, and a lift-off process was used to create a metal mask for etching. The use of a metal mask, instead of a baked e-beam resist mask, was necessary due to the high selectivity ratio required for generating structures only a few microns in height. Finally, the etching process was performed as described above, using an Oxford Plasmalab100 ICP180 CVD/Etch system and a mixture of SF$_6$ and O$_2$ plasma as an etchant. For our process, a SF6:O$_2$ flow ratio of 3:2 produced the best results, with a Si:Ti etching selectivity ratio of approximately 100:1. Increased concentration of O$_2$ in the mixture has two effects: (i) it improves etching anisotropy, which is essential for avoiding shape distortion from the undercut effect, and (ii) it reduces the selectivity ratio, as the Si etch rate gets slower. After etching, the structures were characterized using a Dektak surface profiler and SEM imaging. The micron structure arrays, which contained various shapes (hearts, triangles, pentagons, half-moons and crosses), were fabricated on $1 \times 1$ cm Si chips with inter-structure distances of approximately 5 µm, and structure heights of $5.5 \pm 0.5$ µm (Supplementary Figure 15).

**Micron-sized chamber production and cell growth.** Liquefied agarose (5% w/v) in M9 minimal media (supplemented with 0.2%(w/v) glucose, 0.1%(w/v) casamino acids, 2 µg ml$^{-1}$ thiamine, 40 µM A22 and 20 µg ml$^{-1}$ cephalexin) was dispersed on glass slides and the silica mold (pillar facing downwards) was placed on top. The molds contained either round or rectangular pillars, or various geometrical shapes, as described above. Once the agarose solidified, the mold was removed and ~ 5 µl of live cell culture at OD$_{600}$ 0.4–0.55 (pre-treated with 16 µM A22 for 10–15 min) was applied on top. To allow the cells to adapt to the different shapes, slides were incubated at RT or 30 °C in a parafilm-sealed petri dish together with a wet tissue to prevent drying. After incubation, cells were covered with a pre-cleaned cover glass (♯1.5) for live cell imaging. For STED imaging, cells were first fixed with ice-cold methanol for 5 min and carefully rinsed with PBS prior to cover glass application.

**Microscopy.** Gated STED (gSTED) images were acquired on a Leica TCS SP8 STED 3× system, using a HC PL Apo 100x oil immersion objective with NA 1.40. Fluorophores were excited using a white excitation laser operated at 488 nm for mNeonGreen and 509 nm for mCitrine. A STED depletion laser line was operated at 592 nm, using a detection time-delay of 0.8–1.6 ns for both fluorophores. The total depletion laser intensity was in the order of 20–40 MW cm$^{-2}$ for all STED imaging. The final pixel size was 13 nm and scanning speed was either 400 or 600 Hz. The pinhole size was set to 0.9 AU.

Epi-fluorescence and confocal images were acquired on either a Zeiss LSM780 or Zeiss ELYRA PS1 (both equipped with a $100 \times 1.46$ NA plan Apo oil immersion objective) with acquisition times between 0.3 and 2 s. Time-lapse series for generating kymographs were recorded at 2 s intervals for a time period of at least 118 s.

SIM images were acquired using a Zeiss ELYRA PS1 equipped with a pco.edge sCMOS camera. The final pixel size in SIM images was 24 nm. Individual images were acquired using an acquisition time of 200 ms per image (a total of 15 images were acquired per SIM image reconstruction) and subsequently reconstructed from the raw data using ZEN2012 software. SIM time-lapse movies (containing at least 14 frames) were recorded without time delays between image stacks.

Confocal Z-stacks (focal plane ± ~3.5 µm) were acquired on a Leica TCS SP8 STED 3× system (operated in confocal mode) using predetermined optimal system settings (Leica, LAS X), with 0.22 µm steps (resulting in 30–32 images per stack), and pinhole size 1 AU. All imaging was performed at RT (~ 23–24 °C). In order to

---

### Table 1 Summary of FtsZ cluster dimensions at midcell in various cell shapes

| Cell shape | FP | Drugs[a] | Length (nm) | Width (nm) |
|---|---|---|---|---|
| *FtsZ cluster dimensions* | | | | |
| Circle (WT) | mNG | − | 123 ± 44 | 80 ± 2 |
| Circle (large) | mNG | + | 132 ± 48 | 88 ± 9 |
| Square | mNG | + | 105 ± 40 | 80 ± 18 |
| Square | mCit | + | 118 ± 41 | 86 ± 22 |
| Heart | mNG | + | 129 ± 44 | 84 ± 9 |
| Pentagon | mNG | + | 131 ± 52 | 74 ± 9 |
| Half-moon | mNG | + | 129 ± 45 | 80 ± 10 |
| Star | mNG | + | 140 ± 67 | 76 ± 16 |
| Triangle | mNG | + | 110 ± 35 | 78 ± 11 |
| Cross | mNG | + | 119 ± 21 | 71 ± 18 |

In all cell shapes, the average measured cluster lengths were within 17% of WT, while average widths were within 13%. Numbers represent mean ± S.D. Note that values have been rounded to the nearest integer
FP fluorescent protein, *mNG* mNeonGreen, *mCit* mCitrine
[a]Drugs; A22 [16 µm] and Cephalexin [20 µm]

---

### Table 2 Summary of FtsZ dynamics in various cell shapes

| Cell shape | FP | Drugs[a] | Treadmilling speed (nm s$^{-1}$) | $t_{1/2}$ recovery (s) |
|---|---|---|---|---|
| FtsZ dynamics | | | | |
| Circle (WT) | GFP | − | 26 ± 15 | 8 ± 2 |
| Circle (large) | GFP | + | 30 ± 18 | 8 ± 2 |
| Square | mCit | + | 28 ± 13 | 10 ± 3 |
| Square | GFP | + | 25 ± 11 | 9 ± 3 |
| Heart | mCit | + | 23 ± 10 | 7 ± 1 |

Numbers represent mean ± S.D. Note that values have been rounded to the nearest integer
FP fluorescent protein, *mCit* mCitrine
[a]Drugs; A22 [16 µm] and Cephalexin [20 µm]

confirm that the cells were immobile, and that no visible cell movements occurred in the wells, each cell was initially monitored using brightfield and epi/confocal fluorescence illumination[14]. In this way, we could eliminate the contribution of motion blurring to any observed movements captured during image acquisition.

**FRAP measurements**. Confocal FRAP measurements were performed on a Zeiss LSM780 system using a $100 \times 1.4$ NA plan Apo oil immersion objective and pin-hole size $60\,\mu m$[14]. Bleaching was performed for 0.5–0.7 s using 100% laser power applied over the region of interest. Data were collected in time intervals of 1–2 s until steady state was reached. Following background correction, and to account for overall successive bleaching, the fluorescence intensity (F) of the bleached region (half a ring) was normalized to the average ring fluorescence of an unbleached area of the same size, for each time point (t); $F_{NORM}(t) = F_{BLEACHED}(t) * (F_{BLEACHED}(t) + _{UNBLEACHED}(t))^{-1}$. All data were exported to Origin9 Pro and data points were fitted to the single exponential function $F(t) = F_{end} - (F_{end} - F_{start}) * e^{-kt}$, where $F(t)$ is the fluorescence intensity at time $t$, $F_{end}$ is the fluorescence intensity at maximum recovery, $F_{start}$ is the fluorescence recovery momentarily after bleaching (at $t = 0$), and $k$ is a free parameter. The recovery half-time was then extracted from $t_{1/2} = \ln 2 * k^{-1}$. Importantly, all cells were scanned from top to bottom in order to find the division plane (in which the rings reside).

**Image analysis**. Image analysis was performed using Fiji. When necessary, images were background-corrected using a rolling ball with radius 36. Image stacks were motion-corrected using the plug-in StackReg. Kymographs were generated from time-lapse images using the KymoResliceWide plugin (line width 5), from which treadmilling speeds were calculated using the slope of the fluorescence trace[21].

STED images were deconvolved using Huygens Professional deconvolution software (SVI, the Netherlands). FtsZ-ring diameters were extracted from the average values of the Gaussian fitted fluorescence profiles drawn from 12 to 6 o'clock and 3 to 9 o'clock. Side lengths of shaped cells were determined by applying line profiles in ImageJ. The lengths and widths of individual FtsZ clusters were obtained using line scans (line size 4) over at least five randomly selected individual fluorescence spots from each deconvolved cell image, whereby a Gaussian was fitted to the intensity profiles in order to extract the full width at half maximum (FWHM). Note that the long and short axes of each individual FtsZ cluster were assigned as length and width, respectively, regardless of orientation relative to the membrane. FtsZ cluster dimensions are given in mean ± S.D. $n$ indicates number of cells, unless explicitly specified.

**Statistical analysis**. For statistical analyses, two-tailed Student's $t$-tests were performed using Origin Pro 9. A $p$-value of $<0.05$ was considered as statistically significant.

## Data availability
The data supporting the findings of the study are available in this article and its Supplementary Information files, or from the corresponding author upon request.

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

## Acknowledgements

The authors would like to thank Prof. Harold Erickson (Duke Uni.) for sharing the FtsZ-mNeonGreen strain. Prof. Daniel Daley (Stockholm Uni.) is acknowledged for valuable suggestions to improve the manuscript. BS is supported by JSPS KAKENHI (grant number JP17K15694). Work in the SCB unit at OIST is funded by core subsidy from Okinawa Institute of Science and Technology Graduate University.

## Author contributions

B.S. conceived the study and performed the experiments. A.B. and B.S. designed and engineered the micron pillar arrays. H.C. contributed reagents. B.S. and U.S. analyzed the data. B.S. wrote the manuscript with input from all authors. All authors approved the final version of the manuscript.

## Additional information

**Competing interests:** The authors declare no competing interests.

