## [Peer Review File · Nature Communications]

Reviewers' comments:

Reviewer #1 (Remarks to the Author):

In this manuscript, Soderstrom et al. synthetically remodel *E. coli* cells into various large, geometric shapes to probe whether cell shape alters FtsZ assembly and dynamics using super resolution microscopy (STED, SIM). As I am unaware of any literature suggesting FtsZ localization is specifically shape dependent, (e.g. it localizes to negative curvature), not unexpectedly they find FtsZ "patches" retaining normal dimensions and treadmilling speeds across the shapes tested, suggesting that its assembly is largely shape-independent. The work is novel, but I am unsure precisely what the specific contribution of the work is to understanding of FtsZ behavior and function in normal cells. Critically, the authors offer little justification or context as to why they believed geometric constraints on the membrane might affect properties of the Z ring in the first place beyond noting briefly that *in vitro* FtsZ filaments self-organize into a structure approximately the width of a cell. Alone, this piece of evidence does not seem like sufficient basis for the study, considering the conditions for FtsZ assembly and activity at the membrane are substantially different *in vivo* (e.g. FtsZ is tethered to the inner membrane by FtsA's amphipathic helix; FtsZ regulators may constrain "patch" dimensions, etc.). Without more justification and support for their approach, the manuscript risks being viewed as a technique seeking a question and is unlikely to be notable to the journal's broad audience or by experts in the field.

Additional comments:

1. Lines 65-67: This is a significant overstatement. While the authors find that geometric properties of the membrane do not seem to affect Z ring properties in their synthetic system, other membrane properties (e.g. lipid composition, membrane-tethering proteins) may still affect FtsZ.
2. Lines 119-120: This, again, is an assumption. Other mid-cell localizing systems may play a role in this (e.g. *slmA*-mediated nucleoid occlusion, *ter*). If the authors indeed believe the *min* system is functional in these cells and leading to correct mid-cell Z ring formation, they should show it directly.
3. Lines 127-130: Did the authors identify cells that did not have "normal" Z rings, and if so, what was the frequency of these events across different shapes?
4. Given their conclusion that FtsZ assembly is not affected by the cell shape, it is surprising that the authors did not present data on treadmilling speed for the FtsZ in the square cells since they performed this measurement for all the other shapes.

Reviewer #2 (Remarks to the Author):

In "Cell shape independent FtsZ dynamics in synthetically remodeled cells", Söderström and colleagues describe the structure and dynamic behavior of FtsZ filament clusters in *E. coli* cells of different shapes and dimensions. The key take-home of the study is that the assembly properties of FtsZ are remarkably robust to changes in cell/membrane geometry. The manuscript is straightforward, clear, interesting, and fun to read.

My only experimental comment is to ask whether the authors measured/observed clusters that treadmill through/across regions with large changes in cell geometry or angles, i.e. the sharp corners of square cells or the points of the heart-shaped cells? The observation that some heart-shaped cells are only "half-full" might suggest that clusters cannot readily move across regions with severe angles and would have to assemble/nucleate independently on either side of these

regions. Did they observe clusters that treadmill continuously across these regions or, conversely, observe that clusters move to, but not through, these areas?

Other points:

1. Line 66 - "...underlying membrane environment is not a deciding factor for FtsZ filament maintenance..." In this case "environment" is too broad since it could include chemistry/biochemistry of the membrane as well. Something like "geometry" or "shape" would be better.
2. Lines 73-75 - "At the heart of this process is..." The details included in this statement apply to E. coli and its closest relatives (i.e. ZipA is not broadly conserved), so "in E. coli" should be stated somewhere.
3. Throughout, the authors refer to "sculptured" cells. This phrasing was a bit odd to me. "Sculpted" or "shaped" would be better.
4. Lines 109-110 - "We wanted to see if this was also true for cells growing in the opposite direction..." What is meant by "in the opposite direction"? Consider rephrasing for clarity.
5. Line 132 and elsewhere - Is FtsZ cluster width determined in the radial or longitudinal direction? I assume radial since they're viewing the ring in cross-section, but this should be explicitly stated.
6. Lines 162-164 - "Surprisingly, the average $t_{1/2}$ recovery time was the same for Z-rings with a wide range of diameters." Why is this surprising? The authors already demonstrated that the treadmill rate (which was previously shown to be linked to GTPase rate) was the same, so assuming that [FtsZ] scales with volume, why would the turnover rate be expected to be different at different diameters?
7. Line 187 and throughout - "This suggests that FtsZ filament dimensions in vivo..." The authors cannot conclude anything about "filament" dimensions or dynamics, only "cluster" dimensions, since even with super-resolution light microscopy methods individual filaments/polymers are not resolved.
8. Line 203 - "sharped-cornered" should be "sharp-cornered"
9. Figure 1b - The graphic for the experimental set-up would be improved if it included the pillars used to capture cells standing up.
10. Figure 1g - Why is the width for untreated cells not included?
11. Line 686-7 - Add "at the membrane" to the end of this sentence for clarity.
12. Lines 708-9 - Add units (nm) to the measurements of FtsZ cluster length and width.
13. Figure 4f - The 3D graph is difficult to view. Consider presenting the data in 2D (perhaps including the data for only one or the other fluorophore in the main figure and moving the other to the supplement).
14. Lines 311-314 - This sentence is not entirely clear. Are the authors suggesting that they're showing for the first time that FtsZ dynamics are independent of divisome function in driving constriction (i.e. PG synthesis)? This was shown previously in the original papers describing FtsZ treadmilling.

Reviewer #3 (Remarks to the Author):

The manuscript Cell shape independent FtsZ dynamics in synthetically remodeled cells by Bill Söderström et al. implemented their novel micron pillar design to study the FtsZ dynamics in live E. coli cells with artificial shapes. Using drug treatment the authors morphed E. coli cells into a variety of shapes with drastically different perimeters and curvatures. They found that the treadmill rate, FRAP rate, and dimensions of FtsZ clusters/densities remained constant in all types of cell shapes and sizes, suggesting that these dynamics of FtsZ is an intrinsic property of FtsZ polymerization but not influenced by other factors. These results will be of interest to the bacterial cell division and cytoskeleton community, and the new method of using micron pillars to will be useful for the community to study bacterial cell shape and size. To further improve the work, I suggest the authors address the following comments:

1. The authors attributed the same FRAP rate in larger cells to higher number of FtsZ filaments (line 164 and Figure 2c). This may not necessarily be true-- the FRAP recovery time constant (τ) should be independent of FtsZ concentration (fluorescence) but only related to the rate constants of the exchange reaction.
2. The authors should provide quantification either by single cell fluorescence or ensemble Western to verify FtsZ concentrations in cells with altered shapes. These measurements will provide additional evidence relating to the formation of Z-shapes with the amount of available FtsZ filaments. For example, square cells with ~50 times more volume than WT cells would dilute the FtsZ concentration below the critical concentration for polymerization if there is no upregulation of FtsZ expression. How could they form filaments and eventually a ring?
3. The Min system may oscillate in different patterns in cells of altered shapes and sizes. Could the authors comment on how the Min system would affect FtsZ dynamics here?
4. The authors should investigate more quantitatively on the relationship between FtsZ treadmilling rate/polymerization and membrane curvature. From the few representative images it looks like that at negative curvatures (in heart- and star-shaped cells where the cytoplasmic angle of the two adjacent membranes is > 180 degree) there appeared to be little FtsZ densities. The authors could perform minimally an autocorrelation analysis of FtsZ density/dynamics with curvature in order to draw the conclusion on the influence of curvature on FtsZ distribution and dynamics convincingly.
5. Since the dynamics of FtsZ filaments is similar with bigger or unnatural curvature, this result supports previous work that FtsZ should not be the major force generator for constriction. The authors could discuss this supporting evidence in the context of the long-lasting debate of constriction force generation.
6. How do the authors exclude the possibility that cells in big micro holes rotate or move up and down? Can a fixed cell experiment be done to demonstrate?

Some minor comments:

1. The scale bars in Figure 1c to k are different (between SIM and STED). Is there any specific purpose? Otherwise, it is misleading.
2. The definition of FtsZ 'densities' is misleading-- it could be FtsZ ring concentrations (number of FtsZ molecules divided by the volume of the ring). I think the authors meant FtsZ clusters or filaments reported in other super-res imaging studies.

Reviewers' comments:

Reviewer #1 (Remarks to the Author):

In this manuscript, Soderstrom et al. synthetically remodel *E. coli* cells into various large, geometric shapes to probe whether cell shape alters FtsZ assembly and dynamics using super resolution microscopy (STED, SIM). As I am unaware of any literature suggesting FtsZ localization is specifically shape dependent, (e.g. it localizes to negative curvature), not unexpectedly they find FtsZ “patches” retaining normal dimensions and treadmilling speeds across the shapes tested, suggesting that its assembly is largely shape-independent. The work is novel, but I am unsure precisely what the specific contribution of the work is to understanding of FtsZ behavior and function in normal cells. Critically, the authors offer little justification or context as to why they believed geometric constraints on the membrane might affect properties of the Z ring in the first place beyond noting briefly that *in vitro* FtsZ filaments self-organize into a structure approximately the width of a cell. Alone, this piece of evidence does not seem like sufficient basis for the study, considering the conditions for FtsZ assembly and activity at the membrane are substantially different *in vivo* (e.g. FtsZ is tethered to the inner membrane by FtsA’s amphipathic helix; FtsZ regulators may constrain “patch” dimensions, etc.). Without more justification and support for their approach, the manuscript risks being viewed as a technique seeking a question and is unlikely to be notable to the journal’s broad audience or by experts in the field.

AU: First of all, we would like to thank the reviewer for taking the time to review our work and provide constructive criticism. It is appreciated!

To address the points made:

We have provided stronger justification for the study and further clarified why this study is of importance for better understanding the general behavior of FtsZ in a cellular context (Lines 91-96). This work shows for the first time, and in a systematic manner, how robust the Z-ring is, as it is able to form and maintain its dynamics in a broad range of unnatural shapes. In addition, the overall shape independent dynamics also support the growing body of evidence that FtsZ is not the major force generator in constricting the cell envelope. We have included extra text in the discussion clarifying this (Lines 336-339).

Additional comments:

1. Lines 65-67: This is a significant overstatement. While the authors find that geometric properties of the membrane do not seem to affect Z ring properties in their synthetic system, other membrane properties (e.g. lipid composition, membrane-tethering proteins) may still affect FtsZ.

AU: We agree with the reviewer that this was an overstatement. We have revised the sentence accordingly. Lines 65-66.

2. Lines 119-120: This, again, is an assumption. Other mid-cell localizing systems may play a role in this (e.g. *slmA*-mediated nucleoid occlusion, *ter*). If the authors indeed believe the *min* system is functional in these cells and leading to correct mid-cell Z ring formation, they should show it directly.

AU: We agree with the reviewer that this may have been an assumption. We have changed the text to better reflect the rationale behind the statement.
Lines 124-126 and 129-131.

3. Lines 127-130: Did the authors identify cells that did not have “normal” Z rings, and if so, what was the frequency of these events across different shapes?

AU: In order to avoid misunderstanding, we have changed the text to clarify that essentially all Z-rings looked normal. Lines 136-137.

4. Given their conclusion that FtsZ assembly is not affected by the cell shape, it is surprising that the authors did not present data on treadmilling speed for the FtsZ in the square cells since they performed this measurement for all the other shapes.

AU: We hope that we did not misunderstand the reviewer's question, but we were surprised by this comment, as a large part of Figure 4 (Fig. 4a-d) shows treadmilling data and quantification of the speed in squares. We have rearranged the figure to better clarify that panels b and c show example kymographs taken over corners in square-shaped cells. Neither the continuity nor speed of FtsZ treadmilling was affected by the square shape of the cells. Quantification of treadmilling speeds is presented in panel d. Additional text explaining this has also been added at lines 227-229 and 744-745.

Reviewer #2 (Remarks to the Author):

In "Cell shape independent FtsZ dynamics in synthetically remodeled cells", Söderström and colleagues describe the structure and dynamic behavior of FtsZ filament clusters in *E. coli* cells of different shapes and dimensions. The key take-home of the study is that the assembly properties of FtsZ are remarkably robust to changes in cell/membrane geometry. The manuscript is straightforward, clear, interesting, and fun to read.

My only experimental comment is to ask whether the authors measured/observed clusters that treadmill through/across regions with large changes in cell geometry or angles, i.e. the sharp corners of square cells or the points of the heart-shaped cells? The observation that some heart-shaped cells are only "half-full" might suggest that clusters cannot readily move across regions with severe angles and would have to assemble/nucleate independently on either side of these regions. Did they observe clusters that treadmill continuously across these regions or, conversely, observe that clusters move to, but not through, these areas?

AU: First of all, we would like to thank the reviewer for taking the time to review our work and provide constructive criticism. It is appreciated!

To address the point made:

We realized that we did not present this piece of information well enough in the first version of the manuscript. We have rearranged Figure 4 to better reflect that FtsZ treadmilling is not affected by the square shape of the cells. Figure 4a-d show treadmilling in square-shaped cells. Specifically, Figures 4b and 4c (kymographs) show that FtsZ treadmilling does not seem to be affected by the presence of a sharp corner. We have also added new analysis that shows treadmilling over cell regions with severe angles (e.g. hearts and pentagons).

We have included new text clarifying this at lines 226-228, in the revised Figure 4 (and lines 744-745), in a new supplementary figure (Figure S14) and at lines 288-292 and 320-321.

Other points:

1. Line 66 - "...underlying membrane environment is not a deciding factor for FtsZ filament maintenance..." In this case "environment" is too broad since it could include chemistry/biochemistry of the membrane as well. Something like "geometry" or "shape" would be better.

AU: We have replaced the word "environment" with "geometry".

2. Lines 73-75 - "At the heart of this process is..." The details included in this statement apply to *E. coli* and its closest relatives (i.e. ZipA is not broadly conserved), so "in *E. coli*" should be stated somewhere.

AU: We thank the reviewer for pointing this out. We have now specified that we are specifically referring to *E. coli*. Line 74.

3. Throughout, the authors refer to "sculptured" cells. This phrasing was a bit odd to me. "Sculpted" or "shaped" would be better.

AU: We have changed "sculptured" to "sculpted" throughout the manuscript.

4. Lines 109-110 - "We wanted to see if this was also true for cells growing in the opposite direction..." What is meant by "in the opposite direction"? Consider rephrasing for clarity.

AU: We have clarified the sentence. Line 115.

5. Line 132 and elsewhere - Is FtsZ cluster width determined in the radial or longitudinal direction? I assume radial since they're viewing the ring in cross-section, but this should be explicitly stated.

AU: We have clarified that the widths were measured as "radial widths" throughout the manuscript.

6. Lines 162-164 - "Surprisingly, the average $t_{1/2}$ recovery time was the same for Z-rings with a wide range of diameters." Why is this surprising? The authors already demonstrated that the treadmill rate (which was previously shown to be linked to GTPase rate) was the same, so assuming that [FtsZ] scales with volume, why would the turnover rate be expected to be different at different diameters?

AU: We agree with the reviewer that "surprisingly" was an ill-chosen word. We have changed the sentence. Line 175.

7. Line 187 and throughout - "This suggests that FtsZ filament dimensions in vivo..." The authors cannot conclude anything about "filament" dimensions or dynamics, only "cluster" dimensions, since even with super-resolution light microscopy methods individual filaments/polymers are not resolved.

AU: We thank the reviewer for pointing out this mistake, and we have changed the word from "filament" to "cluster".

8. Line 203 - "shaped-cornered" should be "sharp-cornered"

AU: Changed.

9. Figure 1b - The graphic for the experimental set-up would be improved if it included the pillars used to capture cells standing up.

AU: Here we respectfully disagree with the reviewer and believe that the aesthetics of the figure is best kept as is. Example of micron pillars are found in the supplementary information.

10. Figure 1g - Why is the width for untreated cells not included?

AU: We apologize for the oversight. We have now included the widths for untreated cells in the figure.

11. Line 686-7 - Add "at the membrane" to the end of this sentence for clarity.

AU: This is no longer applicable, as we have removed that part of the figure.

12. Lines 708-9 - Add units (nm) to the measurements of FtsZ cluster length and width.

AU: Done.

13. Figure 4f - The 3D graph is difficult to view. Consider presenting the data in 2D (perhaps including the data for only one or the other fluorophore in the main figure and moving the other to the supplement).

AU: We have changed the 3D graph to a standard 2D bar graph.

14. Lines 311-314 - This sentence is not entirely clear. Are the authors suggesting that they're showing for the first time that FtsZ dynamics are independent of divisome function in driving constriction (i.e. PG synthesis)? This was shown previously in the original papers describing FtsZ treadmilling.

AU: We have changed and clarified the sentence. We do not suggest that we for the first time show that FtsZ dynamics are independent of divisome function, but rather, that we provide additional evidence for this, and support the idea that FtsZ may not be the main force generator during constriction in rod shaped bacteria, consistent with what has been shown previously in cocci (Monteiro *et al.* Nature 2018). Lines 336-339.

Reviewer #3 (Remarks to the Author):

The manuscript Cell shape independent FtsZ dynamics in synthetically remodeled cells by Bill Söderström et al. implemented their novel micron pillar design to study the FtsZ dynamics in live *E. coli* cells with artificial shapes. Using drug treatment the authors morphed *E. coli* cells into a variety of shapes with drastically different perimeters and curvatures. They found that the treadmilling rate, FRAP rate, and dimensions of FtsZ clusters/densities remained constant in all types of cell shapes and sizes, suggesting that these dynamics of FtsZ is an intrinsic property of FtsZ polymerization but not influenced by other factors. These results will be of interest to the bacterial cell division and cytoskeleton community, and the new method of using micron pillars to will be useful for the community to study bacterial cell shape and size.

AU: First of all, we would like to thank the reviewer for taking the time to review our work and provide constructive criticism. It is appreciated!

To further improve the work, I suggest the authors address the following comments:

1. The authors attributed the same FRAP rate in larger cells to higher number of FtsZ filaments (line 164 and Figure 2c). This may not necessarily be true-- the FRAP recovery time constant (τ) should be independent of FtsZ concentration (fluorescence) but only related to the rate constants of the exchange reaction.

AU: We apologize for the confusion. We have clarified this to show that we believe that the exchange rate of FtsZ subunits in individual filaments is independent of cell size. Lines 176-178.

2. The authors should provide quantification either by single cell fluorescence or ensemble Western to verify FtsZ concentrations in cells with altered shapes. These measurements will provide additional evidence relating to the formation of Z-shapes with the amount of available FtsZ filaments. For example, square cells with ~50 times more volume than WT cells would dilute the FtsZ concentration below the critical concentration for polymerization if there is no upregulation of FtsZ expression. How could they form filaments and eventually a ring?

AU: It was not technically possible to extract sculpted cells from the agarose matrix in a controlled and reproducible manner, and therefore we were not able to perform any quantitative Western blotting to quantify FtsZ levels in sculpted cells.

However, we would like to point out that we have used a strain of *E. coli* in which the only copy of FtsZ is a fluorescent FtsZ-mNeonGreen, and that this strain produced normal-looking (albeit bigger) FtsZ rings in both large cells and square-shaped cells, indicating that the cells possibly upregulated FtsZ expression to compensate for the larger cell volume.

Furthermore, per the reviewer's request, we performed single cell fluorescence intensity measurements over the division plane for a number of different cell sizes, and found that the intensity levels increased with cell perimeter (Supplementary Figure S5).

We have included new text discussing the possibility that larger cells may upregulate FtsZ expression at lines 136-141.

3. The Min system may oscillate in different patterns in cells of altered shapes and sizes. Could the authors comment on how the Min system would affect FtsZ dynamics here?

AU: The wells were designed so that cells would only have pole-to-pole oscillations based on the dimensions calculated in the paper "Symmetry and scale orient Min proteins patterns in shaped bacterial sculptures" by Fu *et al.* (Nature Nanotechnology 2015). We have commented on this in lines 129-131.

4. The authors should investigate more quantitatively on the relationship between FtsZ treadmilling rate/polymerization and membrane curvature. From the few representative images it looks like that at negative curvatures (in heart- and star-shaped cells where the cytoplasmic angle of the two adjacent membranes is > 180 degree) there appeared to be little FtsZ densities. The authors could perform minimally an autocorrelation analysis of FtsZ density/dynamics with curvature in order to draw the conclusion on the influence of curvature on FtsZ distribution and dynamics convincingly.

AU: We have clarified that FtsZ treadmilling was continuous over sharp corners in squared cells (Lines 227-229 and Figure 4), and we have also included new analyses on other cells shapes (e.g. hearts and pentagons) containing sharp-angled regions, indicating no difference or change in treadmilling speeds over these regions. Lines 288-292 and Supplementary Figure S14.

5. Since the dynamics of FtsZ filaments is similar with bigger or unnatural curvature, this result supports previous work that FtsZ should not be the major force generator for constriction. The authors could discuss this supporting evidence in the context of the long-lasting debate of constriction force generation.

AU: We thank the reviewer for pointing this out to us, and have subsequently added explanatory text in the discussion. Lines 336-339.

6. How do the authors exclude the possibility that cells in big micro holes rotate or move up and down? Can a fixed cell experiment be done to demonstrate?

AU: We understand the reviewer's concern, and have therefore added information in the methods section to clarify that, before imaging, each cell was observed by brightfield and epi illumination to make sure no visible movements were observed. If anything, fewer movements were observed for larger cells, probably because they expand radially and push against the walls of the agarose wells, creating more friction. Lines 455-460.

To eliminate any doubts about movement or rotation of cells in the holes, we also performed FRAP measurements on fixed cells expressing FtsZ-GFP with various radii in big holes. These additional measurements show that cells were immobile and not rotating, even in big holes (Supplementary Figure 8).

Some minor comments:

1. The scale bars in Figure 1c to k are different (between SIM and STED). Is there any specific purpose? Otherwise, it is misleading.

AU: The scale bar indicates the same length (1 μm) in all images, except in the close-up image 1f (bar = 0.5 μm), where the 1 μm scale bar did not fit well into that image.

2. The definition of FtsZ 'densities' is misleading-- it could be FtsZ ring concentrations (number of FtsZ molecules divided by the volume of the ring). I think the authors meant FtsZ clusters or filaments reported in other super-res imaging studies.

AU: We thank the reviewer for pointing out this mistake, and we have exchanged "densities" for "clusters" throughout the manuscript.

REVIEWERS' COMMENTS:

Reviewer #1 (Remarks to the Author):

The authors have done a nice job of responding to reviewer comments. I have no further concerns.

Reviewer #2 (Remarks to the Author):

The authors have addressed my major concerns. I have only a few small comments:

Lines 88-95. An additional precedent for addressing a possible role of membrane geometry on FtsZ organization is the example of MreB, which has been shown by several groups to respond to both micro- and macro-scale differences in cell/membrane geometry.

Lines 303-306: The Schwille group recently demonstrated that E coli FtsZ can form ring-link vortices in vitro even when tethered by an MTS (i.e. vortices do not depend on FtsA) (Ramirez-Diaz, PLOS Biology 2018). This reference should be added and the reference to FtsA should be removed.

Lines 333-339: I still do not think these conclusions are supported by the present study (the sentence is also quite long and still lacks clarity). The authors have not addressed force generation or divisome assembly/organization and I don't follow how these would be natural implications of their observations. I would suggest just removing this sentence.

Point-by-point responses to reviewers comments and Editorial requests for "Cell shape independent FtsZ dynamics in synthetically remodeled cells", NCOMMS-18-18537A.

Reviewer #1 (Remarks to the Author):

The authors have done a nice job of responding to reviewer comments. I have no further concerns.

AU: We greatly appreciate that the reviewer took the time to evaluate our manuscript a second time.

Reviewer #2 (Remarks to the Author):

The authors have addressed my major concerns.

AU: We wish to thank the reviewer for taking the time to evaluate our manuscript a second time. And we are happy to hear that we have addressed any major concerns raised by the reviewer at an earlier stage.

I have only a few small comments:

Lines 88-95. An additional precedent for addressing a possible role of membrane geometry on FtsZ organization is the example of MreB, which has been shown by several groups to respond to both micro- and macro-scale differences in cell/membrane geometry.

AU: We have added text referring to the possible role of MreB in this context. Lines 82-84 and refs 18, 19 and 20.

Lines 303-306: The Schwille group recently demonstrated that E coli FtsZ can form ring-link vortices in vitro even when tethered by an MTS (i.e. vortices do not depend on FtsA) (Ramirez-Diaz, PLOS Biology 2018). This reference should be added and the reference to FtsA should be removed.

AU: We have added the reference as suggested. But we also retained the FtsA reference, as it is an important early step in our general understanding of FtsZ formation on lipid bilayers. Lines 299-300, and ref 43.

Lines 333-339: I still do not think these conclusions are supported by the present study (the sentence is also quite long and still lacks clarity). The authors have not addressed force generation or divisive assembly/organization and I don't follow how these would be natural implications of their observations. I would suggest just removing this sentence.

AU: We have shorten the sentence and divided it into two. Further did we take away part of the sentence as suggested by the reviewer. Lines 328-333.